# Proteomic and Transcriptomic Analyses to Decipher the Chitinolytic Response of *Jeongeupia* spp.

**DOI:** 10.3390/md21080448

**Published:** 2023-08-15

**Authors:** Nathanael D. Arnold, Daniel Garbe, Thomas B. Brück

**Affiliations:** TUM School of Natural Sciences, Department of Chemistry, Technical University of Munich, Werner-Siemens Chair for Synthetic Biotechnology (WSSB), Lichtenbergstr. 4, 85748 Garching, Germany; nathanael.arnold@tum.de (N.D.A.); daniel.garbe@tum.de (D.G.)

**Keywords:** chitinase, transcriptomics, proteomics, omics, chitin, chitinolytic, glycosidic hydrolase family 18, lytic polysaccharide monooxygenase

## Abstract

In nature, chitin, the most abundant marine biopolymer, does not accumulate due to the action of chitinolytic organisms, whose saccharification systems provide instructional blueprints for effective chitin conversion. Therefore, discovery and deconstruction of chitinolytic machineries and associated enzyme systems are essential for the advancement of biotechnological chitin valorization. Through combined investigation of the chitin-induced secretome with differential proteomic and transcriptomic analyses, a holistic system biology approach has been applied to unravel the chitin response mechanisms in the Gram-negative *Jeongeupia wiesaeckerbachi*. Hereby, the majority of the genome-encoded chitinolytic machinery, consisting of various glycoside hydrolases and a lytic polysaccharide monooxygenase, could be detected extracellularly. Intracellular proteomics revealed a distinct translation pattern with significant upregulation of glucosamine transport, metabolism, and chemotaxis-associated proteins. While the differential transcriptomic results suggested the overall recruitment of more genes during chitin metabolism compared to that of glucose, the detected protein-mRNA correlation was low. As one of the first studies of its kind, the involvement of over 350 unique enzymes and 570 unique genes in the catabolic chitin response of a Gram-negative bacterium could be identified through a three-way systems biology approach. Based on the cumulative data, a holistic model for the chitinolytic machinery of *Jeongeupia* spp. is proposed.

## 1. Introduction

Chitin, the most abundant marine polysaccharide, is composed of β-1,4-glycosidic linked *N*-acetylglucosamine, and to a lesser extent glucosamine monomers. Representing one of the major components of crustacean and insect exoskeletons, radulae of mollusks, and algal and fungal cell walls, 10^9^–10^11^ t are estimated to be biosynthesized annually [1].

In addition to its biodegradability and biocompatibility, chitin and even more so the deacetylated, biologically active form, chitosan, exhibit antimicrobial, antitumoral, and anti-inflammatory properties, rendering them invaluable products for e.g., the biomedical, cosmetic, food, textile, and paper industries [2,3].

With increasing demand for seafood and the rapid growth of its respective industries, crustacean shell waste streams (originating from shrimp, crab, prawn, or lobster fisheries) have a lasting negative impact on ecosystems when disposed of in vast amounts into the ocean or landfills, as commonly practiced [4,5]. Its rigid, crystalline structure renders chitin insoluble, therefore requiring multimodal enzymatic conversion into smaller chitooligosaccharides (COS) to be metabolized by marine or soil organisms. Biological degradation of recalcitrant crustacean shell waste is delayed by calcium carbonate naturally interspersed in between the chitin scaffold, thereby decreasing the surface area for enzymes to act upon, as reflected in the low bioconversion rates of environmental microorganisms [6]. Chitin content and enzymatically relevant physiochemical characteristics, such as the degree of acetylation or solubility, are heavily dependent on the shell waste source material [7]. Crab and lobster exoskeletons naturally contain more calcium carbonate [8,9] in contrast to shrimp [10] or prawns. Furthermore, distinct chitin allomorphs are present, including the predominant and structurally more robust α-isoform, alongside the less common β-isoform found, for instance, in squid pens [11], each requiring different compositions of enzyme cocktails to be hydrolyzed.

Chemical chitin extraction methods are commonly applied on an industrial scale, being both economical and effective [12]. However, they result in hazardous waste streams and unspecific, complex product spectra, which fashion them inapt for high-tech applications and environment-friendly mass production [13].

Therefore, biotechnological extraction methods [14] and enzymatic hydrolysis of shell waste streams are clearly favorable. However, slower conversion rates, higher production costs, and reusability still pose challenges to overcome at an industrial scale, which is why the investigation of novel chitinases is of utter importance.

Vast chitin accumulations in both soil and marine sediments are prevented by chitinolytic organisms, which have evolved sophisticated systems to compete for and exploit the recalcitrant polysaccharide as a carbon and nitrogen source [1]. They are equipped with the enzymatic tools to sense chitin, adhere to it, secrete hydrolases and import the degraded chitooligomers for catabolic assimilation [15]. The chitinolytic machinery—that is all enzymes involved in chitin metabolism—was estimated to comprise up to 100 enzymes and is yet to be understood in its entirety [16,17,18].

In this study, we used our previously isolated and genome-sequenced bacterium *Jeongeupia wiesaeckerbachi* [19], which is closely related to *Jeongeupia naejangsanensis* [20]. Due to the high-resolution genome data available in our group, we chose this microorganism as a model to investigate and characterize its extensive chitinolytic machinery for the first time, using a three-way systems biology approach: First, the most abundant extracellular chitin-active (interacting with chitin or COS molecules) enzymes were identified through LC-MS/MS analysis of the average secretome with colloidal chitin and crab shell chitin as inducers of the chitinolytic enzyme machinery, respectively. Results were critically evaluated by means of combined in silico signal peptide analyses through SignalP 6.0 and SecretomeP 2.0, considering classical and non-classical secretion pathways, respectively. To illuminate the catabolic adaptations in response to chitin intracellularly, comparative proteomic and transcriptomic analyses were conducted. Applying glucose and chitin minimal growth conditions, distinct mRNA and protein expression patterns were revealed through next-generation sequencing and mass spectrometry, respectively, suggesting the involvement of over 600 transcripts and 200 enzymes in the intracellular chitin response of *J. wiesaeckerbachi*. Based on these cumulative results, we propose a simplified holistic model for the chitinolytic machinery of the Gram-negative genus *Jeongeupia* that is exceedingly more complex than previously assumed. The primary gene sequences provided by our synergistic systems biology approach provide the basis for further biochemical enzyme characterizations using a recombinant enzyme production approach.

## 2. Results and Discussion

### 2.1. Extracellular Proteomics

#### 2.1.1. Predicted Chitinolytic Machinery of *J. wiesaeckerbachi* and Subcellular Localization

When cultivated in minimal media with chitin as the exclusive carbon source, *Jeongeupia wiesaeckerbachi* secretes chitinolytic enzymes, which break down the environmental chitin and allow for its import and assimilation. Although this strategy is well described in both fungi and bacteria [21,22], a multitude of export pathways have been identified for the latter, with the Sec and Tat systems noted as major facilitators [23,24,25]. As previously demonstrated [19], the investigated strain’s genome contains 13 glycoside hydrolases of family 18 (GH18) [26,27], which imply not only chitinases (EC 3.2.1.14) of classes III and V but also non-catalytic, accessory proteins. Furthermore, six β-N-acetyl-hexosaminidases, three of which could be attributed to either the GH3 or GH20 family, three GH19 chitinases, and a single lytic polysaccharide monooxygenase (LPMO, AA10; formerly CBM33) are present on a genetic level. An analysis using SignalP 6.0 [28] revealed that only two out of the 13 GH18 did not exhibit an N-terminal signal sequence (gene IDs 635 and 1746), whereas another two signal peptides predictions had comparably lower confidence values of 46% (gene ID 371) and 76% (gene ID 1841), respectively (Appendix A). Regarding the residual putative chitinolytic system, the LPMO, two of the three GH19 (gene IDs 1077 and 302) and merely one of the six β-N-acetyl-hexosaminidases (gene ID 1731) likewise present a leader sequence at their respective N-termini. In conclusion, only 8 out of the 23 enzymes, which form the predicted chitinolytic machinery, do not display a localization signal at all, with two additional inconclusive proteins. Further, all enzymes are predicted to be translocated by the standard secretory Sec-pathway and preprocessed by the leader peptidase I, except for one GH18 (gene ID 2137), which is predicted to be a lipopeptide with a leader peptidase II specific signal peptide instead. In silico analysis using the NetGPI 1.1 glycosylphosphatidyl-inositol anchoring prediction tool revealed no attachment of any of the chitinolytic enzymes to the bacterial cell surface [29].

#### 2.1.2. The Vast Majority of the Chitinolytic System Was Detected Extracellularly

As expected, there was a complete absence of proteins in the culture supernatant when *J. wiesaeckerbachi* was cultured on glucose as a carbon source. This was confirmed by SDS-PAGE and spectrophotometry (results not shown). Due to the absence of proteins, these glucose control samples were rejected as impracticable for the secretome analysis. By contrast, the secreted enzyme amounts were significantly increased when chitin-based substrates were added as an inducing carbon substrate in the medium. Here, protein samples were isolated, purified, and subjected to mass spectrometric-based proteomic analysis. In this instance, all detected peptides were regarded as potentially relevant. To introduce a control mechanism, which might minimize the effects of cell lysis during cultivation and sample preparation, only proteins that were verified in all samples were taken into consideration for subsequent bioinformatic evaluations.

The assessment of the extracellular proteins of three biological *Jeongeupia wieaeckerbachi* replicates in minimal media with colloidal chitin and one sample with unbleached crab shell chitin revealed that peptide fragments of 8 out of the 13 GH18 were abundant extracellularly in all samples (Table 1). If the substrates are regarded separately, a minimum of eleven GH18 were present in the (crab α-chitin derived) colloidal chitin-supplied triplicates’ supernatant, while the unbleached α-chitin from decalcified crab shells only induced the secretion of eight chitinases. This observation is consistent with investigations of the *Cellvibrio japonicus* Ueda107 secretome, which recruits fewer enzymes for α-chitin vs. β-chitin conversion [30]. However, in this study, we utilized two different α-chitin derived substrates with varying degrees of crystallinity. Furthermore, half of the commonly secreted chitinases were among the top 10 most significant proteins, on average, emphasizing their importance in the metabolic response to a chitin-rich environment. Interestingly, both GH18 without a predicted signal peptide were among the commonly secreted enzymes in minimal media with chitin (gene IDs 635 and 1746), indicating a potential non-classical export mechanism [28,31].

Other chitinoplastic (chitin structure-altering) or chitin-active enzymes shared between all secretome replicates involve a putative chitobiose transport system substrate-binding protein (chiE), a FAD-binding oxidoreductase predicted as being AA7 (chitooligosaccharide oxidase (EC 1.1.3.-)) by dbCAN 3.0 [32], a glucosamine kinase, a hexosaminidase, and one lytic polysaccharide monooxygenase (Table 1, lower half). The latter is known to be a crucial, copper-dependent and oxygen-driven auxiliary enzyme for hydrolysis of recalcitrant substrates, such as cellulose and chitin [33,34,35,36,37].

#### 2.1.3. The Lytic Polysaccharide Monooxygenase of the Auxiliary Activity Enzyme Family 10 Seems to Play a Minor Role in α-Chitin Hydrolysis of *Jeongeupia* spp.

Surprisingly, the LPMO’s significance score (or final peptide score, −10logP) provided by the MS/MS peptide identification software PEAKS [38] was rather low, with an average significance rank of 134 throughout all samples. The relatively low LPMO abundance is in concordance with a study by Mekasha et al. [34], which investigated optimal enzyme ratios for a *Serratia marcescens*-based chitin saccharification cocktail. According to the study, 15% of the auxiliary monooxygenase is optimal for shrimp or β-chitins, whereas 2% is ideal when hydrolyzing crab or α-chitins, the latter of which applies to this study’s substrates. Interestingly, the significance score of the LPMO was considerably lower in the unbleached crab shell chitin sample (−10logP of 127.55), compared to those with crab chitin-derived colloidal chitin (−10logP of 245–407). These findings are consistent with the results of the aforementioned study [34], where the LPMO appeared only moderately relevant for α-, as opposed to the generally preferred β-chitin conversion [39]. The exact opposite was reported for chitinase synergy experiments in *Streptomyces griseus*, where the *Sg*LMPO10F exhibited enhanced activity levels on the more stable and crystalline α-chitin over β-chitin [39], resulting in a 30-fold increased substrate solubility. Consequently, the substrate specificity of lytic chitin monooxygenases must be assessed for every enzyme variant. Hereby, the identity of the only surface-protruding aromatic residue in the binding cleft, either Tyrosine (for β-chitin) or Tryptophan (for α-chitin), is reported to influence substrate binding strength [40].

*Jeongeupia wiesaeckerbachi’s* LPMO was more abundant in the amorphous, colloidal chitin-induced secretomes compared to the more crystalline, unbleached crab shell growth conditions. However, no assertions regarding its activity levels can be made, which will be the focus of future work in our lab. Whether the low LPMO abundance in the crab shell-induced secretome is correlated with the enzyme’s low specificity and activity towards α-chitin, or if inhibitory effects of secondary compounds in the unbleached crab shells on chemotaxis and related signal cascades came into effect, remains unclear, but this might prompt relevant questions for industrial applications of unprocessed crustacean waste.

#### 2.1.4. Promising Candidate Proteins for Recombinant Expression Studies

Apart from the obviously chitinoplastic enzymes, a branched-chain amino acid ABC transporter substrate-binding protein (BCAA-ABC-SBP), two TonB-dependent (siderophore) receptors, a class B metal beta-lactamase (MBL) fold metallo-hydrolase, and a porin were present in the top 10 most significant extracellular proteins. Among these, the porin’s function as an outer membrane channel is the most obvious. Knock-out experiments would have to show which (potentially chitinolytic) enzymes would not be present in the secretome anymore and thus are transported through that specific porin. According to domain analysis with InterProScan [41,42], the two as TonB-dependent receptors annotated proteins exhibit large β-barrel domains and might represent ligand gated channels or porins, thus serving as secretion facilitators. The BCAA-ABC-SBP is predicted to have high similarity to the Ile/Leu/Val-binding ABC transporter subunit. It might be upregulated and secreted due to the presence of amido-residues in the environment (media). Unspecific binding to, and import of, *N*′*N*′-diacetylchitobiose is possible [16,43], but a role in chemotaxis, pathogenicity, export, o surface motility cannot be eliminated entirely for the superfamily of ABC transporters [44,45,46]. The involvement of the MBL fold-metallo hydrolase in chitinoplastic activities would have to be studied with knockout or expression experiments since typical functions of this superfamily comprise totally different hydrolytic activities such as alkylsulfatase, as suggested by KEGG annotation and InterProScan results (Table 1).

#### 2.1.5. In Silico Analyses May Aid in Reduction of Cell-Lysis Derived False-Positives

A similar, abovementioned study from Tuveng et al. [30] investigated the secretome of *C. japonicus* on α- and β-chitin-rich biomass with a sophisticated method to ensure cell-free secretomes [47]. With approximately 400 secreted enzymes, depending on the substrate, the secretome was comparable in size to this study, with 386 proteins shared by all samples, although we could not methodically eliminate cell lysis. To compensate for this, we followed the example of Tuveng et al. and conducted an in-silico signal peptide analysis of the putative secretome, thus verifying or challenging their extracellular localization.

In this bioinformatic analysis, we applied the SecretomeP 2.0 and SignalP 6.0 [28,48] software packages, which revealed (Figure 1) that 38% were predicted to be translocated via the classical pathways Sec, TAT, or Pilin, and 12% were predicted to be exported by non-classical pathways, when removing all proteins with a secP score of >0.5 that were also predicted to be secreted classically. In other words, only 50% of the presented minimal chitin-secretome, shared by all samples independent of the supplied chitin form, could be confirmed to be secreted with biocomputational tools, when placing more weight on the SignalP 6.0 algorithm [28] for all five classical secretion pathways over SecretomeP 2.0 [48]. This approach can be justified by the fact that N-terminal secretory signal peptides can be predicted more reliably due to the presence of conserved motifs, which non-classically secreted proteins lack altogether. These are predicted by SecretomeP 2.0 based on specific, pathway-independent protein features instead, including amino acid composition, secondary structure, and degree of predicted structural disorder [48]. Since this represents a more complex task, with substantially less experimentally verified data to feed the neural network, the margin for error in non-classical translocation prediction is naturally higher compared to that of N-terminal signal peptide prediction tools [49].

However, secondary or even tertiary functions of cytosolic or periplasmatic enzymes are described in the literature as so-called moonlighting proteins, which can be accompanied by an unexpected localization inside or outside the cell [50]. For instance, the nucleosome protein histone H1, widely known for its involvement in chromatin structuring, can also function as a thyroglobulin receptor on the outer membrane surface of macrophages [51]. Factoring this phenomenon and potentially unknown secretory mechanisms into the description of the current dataset, probably an excess of 50% of the commonly detected extracellular proteins are exported. Furthermore, while additional analysis with LipoP 1.0 predicted 207/386 of the secretome to be localized in the cytosol [52], it assigned generally low confidence scores of 0.2 for all of these proteins. Of the two previously mentioned GH18 without a signal peptide, one of the two (gene ID 1746) could be confirmed as non-classically exported, hinting at yet-to-be-elucidated translocation pathways that evade our current knowledge and descriptive factors.

We then investigated whether the top 10 most significant proteins found in the secretome (Table 1) were real hits or false positives due to cell lysis events. Deploying the SignalP 6.0 [28] results for classical secretion pathways and our SignalP 6.0-corrected SecretomeP 2.0 [48] prediction results for non-classical secretion pathways (Figure 1), we concluded that all detected extracellular proteins were real hits. Especially the five non-chitin utilization-associated proteins comprising a BCAA-ABC-SBP (gene ID 2871), two TonB-dependent (siderophore) receptors (gene IDs 1064 and 2723), a class B metal beta-lactamase (MBL) fold metallo-hydrolase (gene ID 21), and a porin (gene ID 1471) had to be verified to confirm the validity of the dataset. According to SignalP 6.0 [28], four out of these five proteins are predicted to be secreted by means of the classical SEC pathway. To this end, three proteins (gene IDs 21, 1064, and 1471) were anticipated to be guided outside the bacterial cell with a signal peptide of type I, except for the BCAA-ABC-SBP, being directed by a lipoprotein signal peptide of type II. In contrast, the TonB-dependent siderophore receptor (gene ID 2723) was predicted by SecretomeP 2.0 [48] to be exported non-conventionally. Of the chitin utilization-associated proteins, including four glycosyl hydrolase family 18 proteins (gene IDs 366, 389, 837, and 1746) and the sugar ABC transporter substrate-binding protein (gene ID 441), all proteins were predicted to be exported classically with a signal peptide of type I, except for the non-classically exported GH18 (gene ID 1746), as mentioned above.

#### 2.1.6. Highly Abundant Chitinases Exhibit Two Carbohydrate-Binding Modules

Lorentzen et al. discovered a Gram-negative bacterium in an abandoned ant hill with an unprecedentedly rich chitinase arsenal [53]. During investigation of its secretome, they observed that an increased fraction (93%) of upregulated chitinases contained two carbohydrate-binding modules of the Pfam family 5/12 (CBM5/12). Interestingly, this aligns well with our results, where 3 out of the 4 top 10 most abundant GH18 in the minimal secretome also exhibited one CBM5 and one CBM12, each. For the remaining chitinases or enzymes of hitherto unknown functions, no correlation between the amount of CBM5/12 and abundance could be determined.

### 2.2. Differential Intracellular Protein Expression Using Chitin and Glucose Media

#### 2.2.1. The Intracellular Chitin Response Specializes in Glucosamine Utilization and Cell Maintenance over Hydrolysis

Investigation of the intracellular proteome of *Jeongeupia wiesaeckerbachi* unraveled distinct expression patterns when either chitin or glucose was used as respective carbon sources (Figure 2). A total of 203 putative proteins, depicted in the heatmap or volcano plot, were upregulated at least two-fold with a significance value of 20% and above, corresponding to a *p*-value of <0.05, with chitin.

Strikingly, the five topmost significantly expressed proteins could be linked to chitobiose import, chemotaxis, nitrogen metabolism, and a PrkA family protein serine kinase, the latter of which has been reported to be involved in carbon catabolite repression through mediation of the concomitant signal transduction as well as general stress response [54,55] (Table 2, top).

Closer inspection of the five topmost upregulated proteins in a colloidal chitin-rich environment uncovered a short chain hydrogenase/oxidoreductase [56], an inclusion body family protein, indicating high expression stress, which in turn results in misfolded enzymes [57,58]. Moreover, a substrate-binding protein was detected, which might be related to signal transduction or chemotaxis [59]. Additionally, two proteins of unknown function (Table 2, middle section) could be assigned; these represent promising targets for expression and characterization studies.

Further, glucosamine metabolism-related enzymes, which were upregulated intracellularly by *Jeongeupia wiesaeckerbachi* under chitin induction, included four transport proteins, two hexosaminidases, two polysaccharide deacetylases, and a single chitinase belonging to GH18 (Protein accession number 635), which was also detected extracellularly despite its lack of a signal peptide, indicating a potential moonlighting function or an unconventional translocation pathway [50].

#### 2.2.2. Comparison of Intra- and Extracellular Chitin-Induced Proteomics

The datasets of the significantly and at least two-fold upregulated intracellular proteins and all extracellular proteins in the chitin-rich environment were functionally annotated and classified according to GO-terms with BlastKOALA [60] and subsequently compared (Figure 3). Thereby, approximately 71% of the proteins could be annotated and assigned to the appropriate functional category, while 29% enzymes are of an as-yet unknown function.

The extracellular proteome exhibited a larger fraction of genetic information processing and energy metabolism proteins, whereas the intracellular proteome possessed a higher fraction of environmental information processing proteins.

It is important to note that although the total number of extracellular proteins is higher than that of intracellular proteins in Figure 3, this can be ascribed to the differences in statistical methodology. The intracellular colloidal chitin-induced proteomics data were evaluated differentially to glucose negative controls, with strict statistical thresholds of >2 log fold changes and >20% significance values, whereas all detected extracellular proteins shared among every sample were considered. Expectedly, when inspecting total MS/MS protein detection counts, the number of common intracellular proteins far exceeded that of extracellular proteins with 1475 opposed to 386.

#### 2.2.3. Challenges and Benefits of Biocomputational Approaches

Data evaluation and statistical methodology heavily influence the results of system biology experiments. If the dataset were evaluated with a laxer threshold, considering every intracellular protein to be upregulated 1.2-fold (or 20%) instead of 2-fold (or 100%) for example, a total of 257 proteins instead of 203 could be considered. Subsequently, this would translate to approximately 21% more proteins to be considered as either influenced by or directly involved in the intracellular chitin metabolism of *J. wiesaeckerbachi*, exemplifying the difficulty of bioinformatic data evaluation. Hence, subsequent experimental validation is required to conclude the precise role of individual proteins. However, wet lab approaches are inherently slow and costly. Additionally, when looking at complex systems, such as the chitinolytic machinery, which appears to consist of an interplay of over 200 intracellular proteins, knock-out mutant guided experimental validation, for example, would be unfeasible to achieve in a time- and cost-efficient manner.

### 2.3. Differential Transcriptomics

#### 2.3.1. Distinct Transcription Patterns Highlight the Increased Burden of the Metabolic Chitin Response

Illumina Novaseq 6000-guided cDNA-library sequencing of *Jeongeupia wiesaeckerbachi* culture duplicates in glucose or colloidal chitin minimal media yielded 18.65–21.71 million high quality reads. On average, 93.8% of these were unique reads and 96.8% could be mapped to the provided genome. Please refer to Appendix A for detailed information on the sequencing metrics.

Biocomputational evaluation of the differential transcriptomes revealed distinct transcription patterns in response to the respective carbon sources, as evident in the heatmap (Figure 4A). The volcano plot (Figure 4B) visualization serves to elucidate the three core statements of the dataset, that an increased gene count was upregulated more significantly and at higher expression rates with chitin compared to glucose substrate.

In total, 600 transcripts were upregulated at least 20% or 1.2-fold and with an adjusted *p*-value of <0.1 with chitin in contrast to 468 with glucose as the carbon source. The increased gene recruitment, paired with the extraordinarily abundant chitinolytic machinery of the investigated organism [19], leaves room for speculation regarding whether the *Jeongeupia* genus specializes in chitin as a primary carbon source. In contrast to D-glucose, chitin exhibits a more rigid, less accessible and acetylated structure, which plainly requires more enzymes for degradation, deacetylation, and finally, assimilation. Moreover, terrestrial bacteria compete with fungi for soil nutrients, and co-evolution lead to the development of antagonistic mechanisms, such as antibiotics in fungi and cell-wall targeting chitinases in bacteria [61].

When explicitly looking at the five topmost significantly upregulated transcripts (Table 3), no obviously chitinoplastic genes are listed. Rather, a general NirD/YgiW/YdeI family stress tolerance protein encoding gene, two type IVb pilin encoding genes, an inner membrane FtsX-like permease, and a protein of unknown function were detected. YgiW is known to convey hydrogen-peroxide resistance in *E. coli*, but it functions as a general stress response protein to external stimuli [62]. In *Jeongeupia* spp., it might play a central role in the chitin stress response, rendering it a promising target for knockout experiments.

Type IV pili (T4P) are multifunctional protein filaments, populating the surface of many bacteria and archaea [63]. Through rapid assembly and disassembly, T4P enable twitching motility for directed movement towards substrates upon external stimuli but are also involved in the Type II secretion system, DNA uptake, and biofilm formation [64]. The FtsX-like permease family are predicted transmembrane proteins that can release, for example, lipoproteins from the cytosol to the periplasm in an ATP-dependent manner (refer to UniProt accession P57382). Nonetheless, six out of the thirteen genome-encoded GH18 [19] were upregulated during exposure to colloidal chitin, albeit on surprisingly low ranks, underpinning the intra- and extracellular proteomic results. In addition, two of the three GH19 type chitinases and one of the three GH20 hexosaminidases exhibited increased transcript rates.

#### 2.3.2. Low Protein-mRNA Correlation between the Intracellular Datasets

The inquiry for the top hits shared between the proteomic and transcriptomic datasets revealed intriguingly little correlation between upregulated transcripts and detected protein levels.

Refer to Table A1 in the Appendix B for the extensive evaluation results and Appendix A for the most significantly upregulated genes with glucose.

The discrepancy between mRNA and protein abundancies is a well-reported challenge in the systems biology domain and has been subject of intense scientific discussions [65].

With correlation coefficients of about 0.77 in *E. coli*, where one mRNA molecule corresponds to 10^2^–10^4^ respective protein molecules, transcript concentrations were long thought to be unreliable proxies for the prediction of corresponding protein levels and activities [66,67]. Variations of mRNA levels reflect approximately 29% of variations in cellular protein concentration [68]. Translation of genes into proteins, with mRNAs as mediating templates, is an immensely complex process with a multitude of influencing factors: (1) sequence-based translation efficiency, influenced by codon bias or chromatin structure, (2) translation rate modulation through genetic regulatory elements including feedback repression, (3) highly dissimilar protein turnover rates, which are dependent on the ubiquitin–proteasome pathway and temperature, among other things, (4) protein synthesis delay, (5) protein transport, disconnecting measured transcript and enzyme levels through spatial separation in a given compartment, and (6) transcript measurement noise on a methodological level [65,67,69].

When comparing proteome and transcriptome datasets regarding their validity, proteins are more closely related to the phenotype of a cell as a functional expression of its origin gene. Additionally, they are more robust ex vivo, immune to non-functional mRNA artefacts, and outperform transcriptomics in gene function prediction. Nonetheless, transcript concentrations still offer valuable information about imminent protein biosynthesis requirements of a cell [70].

#### 2.3.3. Chitin Metabolism Transcript Upregulation Is Time Dependent

A similar study from Monge et al. concerning the chitinolytic system of *C. japonicus* revealed strong upregulation of chitin degradation-implicated transcripts [71], occupying the top seven most strongly upregulated gene ranks. According to the authors, the upregulation of chitinoplastic genes was more pronounced in the exponential growth phase than the early stationary phase. The latter finding provides a methodological explanation for the relatively low fold changes and adjusted *p*-value rankings of *J. wiesaeckerbachi’s* chitin conversion-related mRNAs. In this study, cultures for transcriptomic investigation were grown on minimal colloidal chitin medium for three days when the majority of the insoluble substrate particles were hydrolyzed, which interfered with RNA extraction. Furthermore, sufficient biomass and concomitant RNA yields, required by the external RNA-sequencing provider Eurofins Genomics, cannot be achieved earlier under these growth conditions. Therefore, the cells were most likely in the (mid to late) stationary phase when relevant transcripts were already degraded. In *E. coli*, mRNA is degraded within 5–10 min [66], therefore transcript levels resemble the recent transcription activity whereas protein levels reflect the accumulated long-term expression. It is further reported that different proteins have distinct optimal concentrations in the cell, which might have been reached and transcription thereof inhibited at this point in time, since protein residence times often exceed that of a cell life cycle anyway [72,73].

Saito et al. conducted a study on the co-transcriptional regulation of chitinase genes in the genome of *Streptomyces coelicolor* A3(2) with Northern blot hybridization, using labelled anti-sense RNAs [74]. Their results demonstrated that colloidal chitin-induced chitinase transcription reaches a maximum after 4 h, emphasizing the importance of temporal expression patterns. Interestingly, only five of the eight genome-encoded chitinase mRNAs could be detected experimentally in varying concentrations, deploying either colloidal chitin or chitobiose. This is in accordance with the several non-traceable chitinolytic machinery-implicated transcripts or proteins of *J. wiesaeckerbachi*, suggesting either specific substrate dependence, non-functionality, or even superfluity of certain genes. In a follow-up study, Saito et al. elaborated that the multiplicity of chitinases in *Streptomyces* spp. has developed through domain deletion and gene duplication [75], which might be extrapolated to other chitinase-rich genera like *Jeongeupia* [19,76].

Pathways connected with high protein cost, such as the chitinolytic machinery, are tightly regulated by fine-tuned transcriptional programs to not unnecessarily waste cellular energy and resources [77]. A GntR family transcription factor [78], annotated as *N*-acetylglucosamine utilization regulator by the KO database, is located just upstream (gene ID 443) of the chitobiose transport system genes (gene IDs 439–442). It was neither found to be upregulated in our transcriptome, which captured the RNA concentrations after three days, nor in the intracellular proteome within the same time frame. Nonetheless, the chitobiose transport proteins under the control of the *N*-acetylglucosamine utilization regulator were upregulated significantly intracellularly Table 2). Again, this emphasizes the temporal delay between transcription and protein synthesis of genetic regulators and their target proteins, as well as protein longevity, given sufficiently low turnover-rates.

The search for additional transcripts involved in gene regulation uncovered 18 upregulated mRNAs in total (Appendix A), with 6 >2-fold upregulated mRNAs under colloidal chitin growth conditions after three days of cultivation. Among them are two transcriptional regulators of the families Rrf2 and GntR (gene IDs 2638 and 2797), two σ^70^ family RNA polymerase factors (gene IDs 299 and 2681), one anti σ^70^ factor (gene ID 300), and one hypothetical transcription factor (gene ID 237) according to SWISS-MODEL [79], which was also upregulated 33-fold in the intracellular chitin-induced proteome.

## 3. Materials and Methods

### 3.1. Chemicals and Consumables

All chemicals were supplied from Sigma-Aldrich (Darmstadt, Germany), and general consumables were obtained from VWR (Darmstadt, Germany). All necessary buffers and enzymes for next-generation genome sequencing were shipped from Pacific Biosciences (Menlo Park, CA, USA). High molecular weight DNA was extracted with the Quick-DNA™ HMW MagBead Kit from Zymo Research (Freiburg, Germany). HMW gDNA shearing was conducted with g-TUBEs (Covaris, Woburn, MA, USA).

### 3.2. Colloidal Chitin and Media Preparation

Colloidal chitin (CC) was prepared according to Murthy and Bleakley [80] with slight modifications. A total of 20 g of crab shell chitin powder (Sigma-Aldrich) was incrementally added to 150 mL 37% HCl under moderate stirring, increasing the viscosity of the solution. When the viscosity decreased sufficiently, more chitin was carefully added. The slur was then incubated for 2–3 h at room temperature under moderate stirring, evading the formation of foam. Afterwards, the non-viscous, fully dissolved chitin of an intense brown color was slowly poured into 2 L of ice-cold deionized water (diH_2_O) in a 5 L glass beaker and vigorously stirred, rapidly swelling to white colloidal chitin. The solution was incubated overnight at 4 °C without stirring and neutralized the following day through the addition of excessive amounts of diH_2_O and subsequent centrifugation in a Beckman JLA8.1000 rotor for 15 min at 10,000× *g* until a supernatant pH of 5 was reached. CC was harvested and kept in the refrigerator until its utilization for liquid chitinase screening media (CSM) or agar plates. The recipe was adapted and modified from [81,82]: 20 g/L (2% *w*/*v*) CC, 0.7 g/L K_2_HPO_4_, 0.3 g/L KH_2_PO_4_, 0.5 g/L MgSO_4_ · 5H_2_O, 10 mg/L FeSO_4_ · 7H_2_O, and 20 g/L agar (optional), adjusting to pH 6.5 for plates or 7 for liquid medium. After autoclaving, 1 mg/L ZnSO_4_ and MnCl_2_ were added from sterile filtrated stock solutions prior to the pouring of agar plates/inoculation of liquid media.

### 3.3. Bacterial Strains

The previously described chitinolytic bacterium *Jeongeupia wiesaeckerbachi* retrieved from environmental samples was utilized for all experiments; its genome is available on NCBI, under the BioSample accession ID SAMN35557021.

### 3.4. Proteomics

#### 3.4.1. Culture Conditions

I. Precultures

*Jeongeupia wiesaeckerbachi* was streaked out onto CSM-agar (pH 6.5, 2% CC (*w*/*v*)) from axenic cryostocks. The precultures were prepared by placing one colony each into 150 mL baffled shaking flasks holding 20 mL tryptic soy broth. Cultivations were carried out in an incubation shaker (New Brunswick Innova 44, Eppendorf, Hamburg, Germany) at 28 °C and 120 rpm overnight. Cell densities were determined spectrophotometrically, measuring the absorption at 600 nm wavelength in 2 mL cuvettes (Nano Photometer NP80, IMPLEN, Munich, Germany).

II. Intracellular Protein Investigation

The main cultures were prepared in 500 mL baffled shaking flasks holding 50 mL of either CSM (pH 7, 2% CC (*w*/*v*)) or modified CSM with 0.5% (*w*/*v*) glucose and 1% (*w*/*v*) NH_4_Cl instead of colloidal chitin. Sufficient bacterial cell amounts were washed twice in sterile phosphate buffered saline (PBS) prior to media inoculation to an OD600 of 0.05. Cultivation parameters identical to those of the precultures (28 °C, 120 rpm) were selected, with incubation times of one (glucose-fed) or three days (chitin-fed), respectively, to acquire enough cell mass. Both carbon sources (glucose or colloidal chitin) were tested in biological quadruplicates, equating to eight samples in total.

III. Extracellular Protein Investigation

In order to examine the enzymes secreted into the culture medium, 500 mL CSM in 5 L baffled shaking flasks was inoculated with *Jeongeupia wiesaeckerbachi* to an OD600 of 0.05 in biological triplicates. Additionally, one flask was prepared with CSM containing 2% (*w*/*v*) processed crab shell chitin (unbleached) instead of CC as the sole C and N source. After three days at 28 °C and 120 rpm, the cultures were centrifuged at 10,000× *g* for 10 min. The supernatants were sterile filtered with a 0.22 µm syringe filter and concentrated using a tangential flow filter membrane (MWCO 10 kDa; Omega 10K Membrane, Pall Cooperation, New York, NY, USA) and a peristaltic pump (Masterflex P/S Model 910-0025, Thermo Scientific, Menlo Park, CA, USA) to a volume of 10–15 mL. Afterwards, 10 kDa MWCO centrifugal filter units (Centriprep, Merck Millipore, Darmstadt, Germany) were applied to further concentrate the secreted crude enzyme mixes to a final volume of approximately 1 mL per sample. Protein concentrations were measured with a photometer based on 260/280 nm absorption ratios (Nano Photometer NP80, IMPLEN, Munich, Germany). Of these protein extracts, 15 µL were transferred into a new reaction tube, mixed with 5 µL 4 × SDS-sample buffer, and boiled for 5 min at 95 °C.

#### 3.4.2. Whole Cell Protein Extraction

The protocol for protein extract and precipitation was slightly modified from Engelhart-Straub and Cavelius [83]. Bacterial cultures were harvested through centrifugation at 8000× *g* for 10 min, and the media supernatant was discarded. The cells were subsequently washed twice with 5 mL of sterile PBS, resuspended in 600 µL PBS, and transferred to 2 mL micro reaction tubes. Afterwards, cell lysis was induced by horizontal vigorous shaking (Vortex Genie 2, Scientific Industries, Bohemia, NY, USA) for 30 min with fine glass beads, supported by 1:3 (*v*/*v*) Protein Extraction Reagent Type 4 (Sigma-Aldrich, St. Louis, MO, USA). After centrifugation at 14,000× *g* for 30 min, protein precipitation was achieved through the addition of 1:1 (*v*/*v*) 20% trichloroacetic acid in HPLC-grade acetone (*w*/*v*), supplemented with 10 mM DL-1,4-Dithiothreitol (DTT). The samples were vigorously vortexed and then incubated at −20 °C for one hour. Following centrifugation at 14,000× *g* for 10 min at 4 °C, the protein pellets were washed twice with HPLC-grade acetone supplemented with 10 mM DTT and air dried under a sterile bench. Lastly, the protein pellets were resuspended in 450 µL 8 M urea with 10 mM DTT and homogenized with a micro pestle suitable for 2 mL micro reaction tubes. Of this protein extract, 15 µL were transferred into a new reaction tube, mixed with 5 µL 4 × SDS-sample buffer, and boiled for 5 min at 95 °C.

#### 3.4.3. Tryptic In-Gel Digestion and LC-MS/MS Analysis

The extracted proteins from whole cells were resolved by SDS-PAGE and subsequently digested with trypsin. The resulting peptides were then separated by reversed-phase chromatography and detected with a mass spectrometer as described next.

The tryptic in-gel digestion protocol and LC-MS/MS analysis with a timsTOF Pro mass spectrometer, coupled with a NanoElute LC System (Bruker Daltonik GmbH, Bremen, Germany) equipped with an Aurora column (250 × 0.075 mm, 1.6 μm; IonOpticks, Melbourne, Australia), were adapted from Fuchs et al. and Engelhart-Straub and Cavelius [83,84]: One-dimensional 12% SDS PAGEs with short stacking gels were deployed to transfer 10 µL of each whole cell protein extract into the resolving gel matrix. Hereby, it is crucial to leave several empty wells between the different conditions (glucose/chitin) to prevent sample migration. Refer to the Appendix A for the full protocol.

The mobile phase consisted of two solvents for reverse-phase chromatography: (A) 0.1% formic acid and 2% acetonitrile in water and (B) 0.1% formic acid in acetonitrile, which was added linearly with a constant flow rate 0.4 µL/min. Both separation cycles started at 2% of B (*v*/*v*). For the less complex extracellular protein mixtures, a short gradient was carried out: t = 25 min and 17% B (*v*/*v*), t = 27 min and 25% B, t = 30 min and 37% B, with t = 33 min and a hold at 95% B for 10 more minutes. In case of the more complex intracellular protein compositions, a longer separation cycle of 100 min was selected: t = 60 min and 17% B, t = 90 min and 25% B, t = 100 min and 37% B, and t = 110 min and 95% B with a hold at 95% B for 10 more minutes. The oven temperature was kept at a constant 50 °C during measurements.

#### 3.4.4. Bioinformatic Analysis

The PEAKS studio software (v.10.6, build 20201221) was utilized for evaluation of the MS/MS tandem spectrometry data of tryptic digested peptides, deploying the annotated genome of *Jeongeupia wiesaeckerbachi* (BioSample accession ID SAMN35557021) as the reference for protein identification [38]. The following “*Database search*” parameters were applied: a precursor mass of 25 ppm using monoisotopic mass and a fragment ion of 0.05 Da for the error tolerance; trypsin as a digestion enzyme; a maximum of two missed cleavages per peptide; a maximum of three variable PTM (post-translational modification) per peptide and estimation of FDR (false discovery rate) with decoy fusion was allowed. For the protein identification, 1.0% FDR with at least one unique peptide per protein was selected. The intracellular glucose and colloidal chitin sample groups were differentially quantified with PEAKSQ, applying a mass error tolerance of 20.0 ppm, an ion mobility tolerance of 0.05 Da, and a retention time shift tolerance of 6 min (auto detect). The fold change and significance were set to 2 and 20, respectively. All proteins were exported and utilized for manual evaluation and plot creation.

The R-Studio software with the ggplot2 package served as the main tool for the creation of plots, if not stated otherwise [24,25].

The functional characterization of both the secreted and intracellular proteomes was conducted with the browser-based BlastKOALA (KEGG Orthology And Links Annotation) tool on the KEGG server [60], utilizing the taxonomy ID (taxid) 885864 or the option “Prokaryote” since the latest update as of May 2023. Carbohydrate-active enzymes (CAZymes) and carbohydrate-binding modules (CBM) in the proteomes and the transcriptome were predicted with the browser based tool dbCAN 3.0, using the HMMER:dbCAN (E-Value < 10^−15^, coverage > 0.35), DIAMOND: CAZy (E-Value < 10^−102^) and HMMER: dbCAN-sub (E-Value < 10^−15^, coverage > 0.35) options [32].

### 3.5. Differential Transcriptomics

#### 3.5.1. Culture Conditions

Like during the investigation of the intracellular proteins (see 3.4.1, II), the main cultures were prepared in 500 mL baffled shaking flasks holding 50 mL of either CSM (pH 7, 2% CC (*w*/*v*)) or modified CSM, with 0.5% (*w*/*v*) glucose and 1% (*w*/*v*) NH_4_Cl instead of colloidal chitin. Sufficient bacterial cell amounts were washed twice in sterile phosphate buffered saline (PBS) prior to media inoculation to an OD600 of 0.05. Cultivation parameters identical to those of the precultures (28 °C, 120 rpm) were selected, with incubation times of one (glucose-fed) or three days (chitin-fed), respectively, to acquire enough cell mass.

#### 3.5.2. RNA Extraction and Quality Control

Bacterial cells were harvested via centrifugation at 6800× *g* for 10 min. Total cell RNA was isolated according to the recommendations of the SV total RNA Isolation System Kit (Promega, Madison, WI, USA). Purity and quantity of the obtained RNA were assessed per photometer based 260/280 nm absorption ratios (Nano Photometer NP80, IMPLEN, Munich, Germany). Furthermore, quality numbers were evaluated with the Qubit 4 fluorometer and the Qubit RNA IQ Assay-Kit (Thermo Fisher Scientific; Waltham, MA, USA). The experiment was performed in biological triplicates, but only two samples per condition (chitin- or glucose-containing media) were analyzed.

#### 3.5.3. Next Generation Sequencing and Bioinformatic Analysis

The EuroFins Genomics Europe Sequencing GmbH (Constance, Germany) performed rRNA depletion, cDNA library construction, next-generation sequencing with the Illumina NovaSeq platform (6000 S4 PE150 XP mode), and the following bioinformatic analyses. Raw sequencing data were cleansed of rRNA reads with RiboDetector [85]. Adapter trimming, quality filtering, and per-read quality pruning were executed with fastp [86]. High quality reads were aligned to the provided *J. wiesaeckerbachi* genome with STAR [87]. Gene-wise quantification was achieved by evaluating transcriptome alignments by means of the software featureCounts [88]. Differential gene expression analysis between the glucose-fed and chitin-fed sample groups was performed using the R/Bioconductor package edgeR [89]. Variant calling for SNP and InDel assessment was conducted with Sentieon’s HaplotypeCaller [90]. Details on the applied software and command line parameters can be found in the Appendix A.

## 4. Conclusions

### 4.1. Chitin-Metabolism Causes Profound Genetic Changes

Through a combination of intracellular and extracellular proteomic analyses, the involvement of 360 unique enzymes within the chitin metabolism of the Gram-negative bacteria genus *Jeongeupia* could be demonstrated, deepening our understanding of natural chitin saccharification systems. Considering all genes with >2-fold increased differential expression rates and with significance values of >20% (proteomics) or adjusted *p*-values of <0.001 (transcriptomics), respectively, 203 intracellular proteins and 244 transcripts (pseudo gene-adjusted) could be detected. The addition of the 192 extracellular enzymes that were both monitored among all samples and confirmed in silico to be secreted, followed by the removal of redundant hits, produces a total of 577 unique genes that were reliably induced by chitin substrates.

Our previously reported dbCAN 3.0-mediated CAZyme prediction [19,32] disclosed the existence of thirteen GH18, three GH19, three GH20, and a single LPMO (AA10) in the genome of *Jeongeupia wiesaeckerbachi*. Twelve of those thirteen GH18 (all but gene ID 1841), one GH19 chitinase (gene ID 302), one GH3 hexosaminidase (gene ID 1323), and all three GH20 hexosaminidases (gene IDs 269, 306 and 1731), in addition to the LPMO (gene ID 148), could be detected in extracellular proteomic samples on colloidal chitin. Intriguingly, the GH20 were secreted on both crab shell and colloidal chitin substrate, while the GH19 was only detected in colloidal chitin samples. These results were confirmed and elaborated through the detection of two GH19 transcripts (gene IDs 302 and 680) in addition to two GH20 transcripts (gene IDs 269 and 306). Overall, only a single GH19 chitinase (gene ID 1077) and two of the three GH3 hexosaminidase (gene IDs 308 and 872) genes of the previously in silico-predicted chitinolytic machinery could not be experimentally verified to be at least transcribed or translated in an enhanced manner under chitinase-inducing growth conditions. Remarkably, most of the chitinolytic system was secreted, or at least exclusively found extracellularly, whereas only a single GH18 (gene ID 635), two GH3 hexosaminidases (gene IDs 308 and 1323), and a single GH20 hexosaminidase (gene ID 269) could be confirmed in the intracellular proteome.

### 4.2. Potential Role of Redox Enzymes in Chitin Hydrolysis

Despite the LPMO’s well-understood function to promote the efficiency of synergistic chitinases on a crystalline substrate [35,91,92,93,94], there are reports where its activity is uncoupled and may play a role in virulence instead [95,96,97]. Similarly, the *Jeongeupia wiesaeckerbachi* LPMO of family AA10 (gene ID 148) was merely among the top 150 most significant secreted proteins on average and was ranked considerably lower in the crab shell chitin sample (opposed to colloidal chitin), specifically. Notably, two additional predicted AA proteins of families 2 and 7 (gene IDs 1157 and 2996, respectively) were more abundant throughout all the secretome samples, independent of the respective substrate. Aside from that, a third FAD-dependent oxidoreductase (gene ID 1920) was detected extracellularly with amorphous chitin, but not crab chitin, suggesting substrate-specific expression. On an intracellular level, an additional oxidoreductase of the SDR (short-chain dehydrogenases/reductases) family (gene ID 1215) was upregulated 50-fold, indicating a general importance of electron transfer chains for bacterial chitin metabolism. Apart from the redox chemistry catalyzing enzymes and the obvious chitinoplastic GH18/19/20 and LPMOAA10 enzymes, two >8-fold upregulated polysaccharide deacetylases (gene IDs 281 and 437), as detected in the intracellular proteome, could represent promising targets for characterization studies. Since the deacetylated form of chitin, chitosan, is the desired molecule for most industrial applications, a strong interest in the targeted and sustainable enzymatic conversion persists. CAZyme prediction of the collective upregulated proteome and transcriptome (>20% increase) dataset unraveled several additional CBM5/12 exhibiting enzymes, which are hallmarks for chitin-active proteins (Table A2; refer to Appendix A for a comprehensive list). Furthermore, the CBM50 or LysM domains have been reported to be involved in penta-*N*-acetyl-chitopentaose (pentamer of *N*-acetylglucosamines) binding [98]. Lastly, some of the secretome-detected GH23 enzymes are of particular interest, being a family of hydrolases that can comprise chitinase activities (EC 3.2.1.14) [99].

### 4.3. Chitin-Rich Environments Prompt a Multitude of Methyl-Accepting Chemotaxis and Motility Proteins

In *Vibrio cholerae*, a single regulatory noncanonical histidine sensor kinase ChiS was identified to orchestrate the catabolic chitin response [16,100,101]. However, genome-wide protein sequence alignment with ChiS did yield no feasible homologies. With the *Vibrio cholerae* chitinolytic signal transduction cascade as a role model, an inquiry into highly transcribed or translated two-component sensor histidine kinases yielded a noteworthy amount of methyl-accepting chemotaxis proteins (MCP) or aspartate receptors, with high log2 fold changes and significance. MCPs are the most common bacterial receptors and mediate transmembrane signal transduction of environmental cues as part of a multiprotein complex, ultimately leading to chemotaxis towards a more favorable environment [102,103]. Hereby, a correlation between genome complexity, habitat, and the amount of genome-encoded MCPs (gene IDs 602, 1841, 3352, 246, 3077, 2218; upregulated 14-, 5-, 3-, 2-, 3.5-, and 1.5-fold, respectively) was observed [104]. Through propagation of conformational changes over receptor-modulating enzymes like CheR (gene ID 36, 12-fold upregulated) and a receptor-coupling protein CheW (gene ID 1614, 2.5-fold upregulated), MCPs control the sensor histidin kinase CheA (gene ID 138, upregulated 5-fold), which subsequently phosphorylates the flagellar-motor receptors CheV and CheY (genes ID 134 and 135; upregulated 5- and 5.5-fold, respectively) [105,106]. Furthermore, a putative transcription factor (gene ID 237) was upregulated 33-fold with high significance in the intracellular proteome, suggesting a central role in chitin adaptation.

Light microscopy imaging (not shown) revealed dense colloidal chitin particle colonization through bacterial cells, implying chemotaxis and possibly pili-mediated twitching motility, as suggested by the transcriptomics data set. Conceivably, *Jeongeupia wiesaeckerbachi* follows a common strategy to secure the released glucosamines against competing soil bacteria and fungi through the abbreviation of transport routes and immediate substrate-binding and import [15].

### 4.4. Schematic Summary of the Cumulative Systems Biology Approach

Based on our combined omics results, we propose a holistic model for the chitinolytic machinery of *Jeongeupia* spp., trying to factor in the most relevant findings for chitin metabolism (Figure 5). In light of the approximately 550 upregulated genes induced through chitin substrates, the schematic does not claim to be more than a mere approximation of reality. Nevertheless, it does provide an overview of the putatively involved main enzymes and avails to appreciate the complex interplay of gene transcription, protein translation, and signal transduction, which assembles the catabolic chitin response of chitinolytic bacteria.

## Figures and Tables

**Figure 1 marinedrugs-21-00448-f001:**
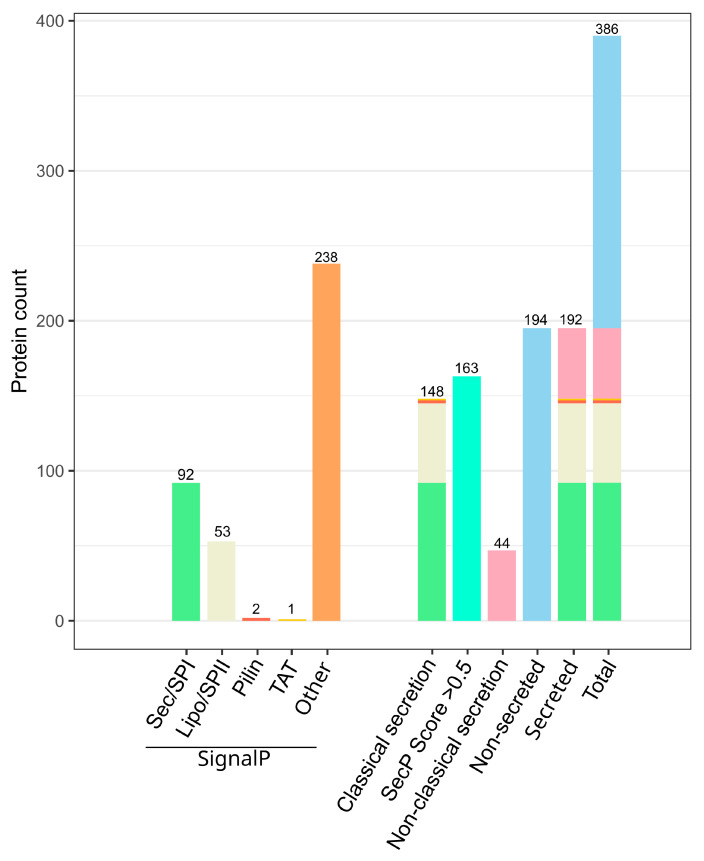
Predicted signal peptides of *Jeongeupia wiesaeckerbachi* in the chitin-induced extracellular proteome. Enzymes predicted to be translocated with the classical secretory pathways Sec, TAT, and Pilin or non-classical pathways (“Other”) by SignalP 6.0 are illustrated on the left side. The sum of all classically secreted enzymes (SignalP), non-classically secreted enzymes predicted by SecretomeP 2.0 with scores of >0.5, and actual non-classically secreted enzymes through comparison of the two algorithms are on the right. Non-secreted enzymes are assembled by the total minimal chitin induced proteome count (386) subtracted by the sum of all classical (148) and non-classical (44) translocated enzymes.

**Figure 2 marinedrugs-21-00448-f002:**
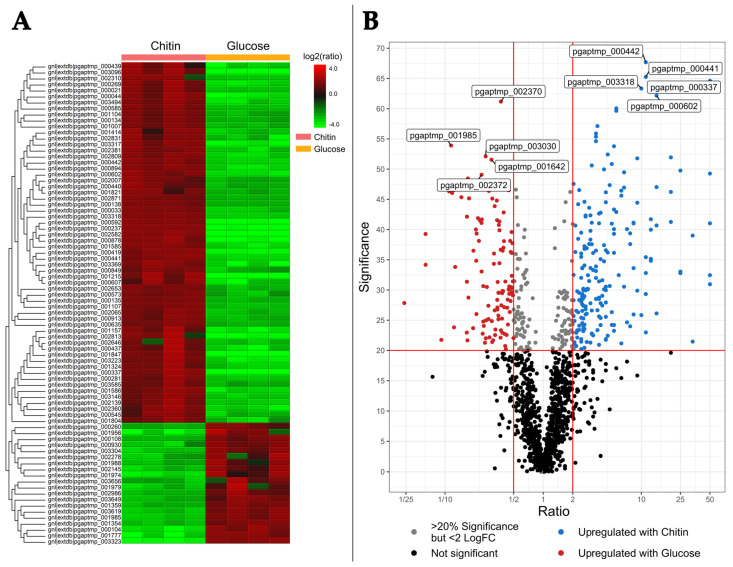
Heatmap (**A**) and volcano plot (**B**) of the differential intracellular proteomic analysis of *Jeongeupia wiesaeckerbachi* with chitin and glucose minimal media. (**A**) The heatmap was generated with PEAKS studios with the following parameters: Fold Change >2, Significance >20, Significance method ANOVA. Upregulated proteins are depicted in red and downregulated proteins in green. (**B**) Volcano plot of the same dataset. Dots in red represent proteins with a fold change >2 and a significance of >20% under glucose media; blue dots represent proteins, which are upregulated under chitin media conditions applying the same statistic thresholds. The top 5 most significant proteins are labelled with their respective gene ID. Refer to Table 2 for detailed information on gene functions.

**Figure 3 marinedrugs-21-00448-f003:**
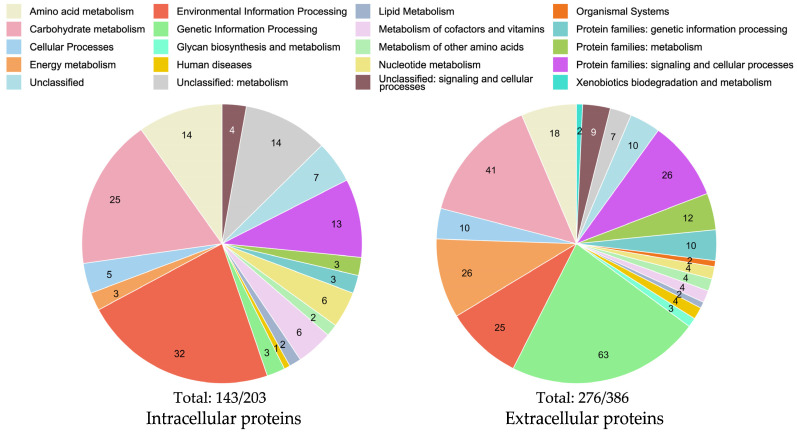
Functional classification of intra- and extracellular proteins of *Jeongeupia wiesaeckerbachi* under chitin conditions according to GO-terms. Both datasets were gathered in quadruplicates. Intracellular proteomics data were evaluated differentially to glucose-supplied cells, with significance values >20% and >2 log2 fold changes; refer to the heatmap (Figure 2). Extracellular proteins depicted were shared amongst all samples. Category counts are shown in each individual pie chart. Totals under the pie charts refer to the annotated fractions of the total protein entries submitted to BlastKOALA, with 203 intracellularly and 386 extracellularly, corresponding to approximately 70% annotated proteins each.

**Figure 4 marinedrugs-21-00448-f004:**
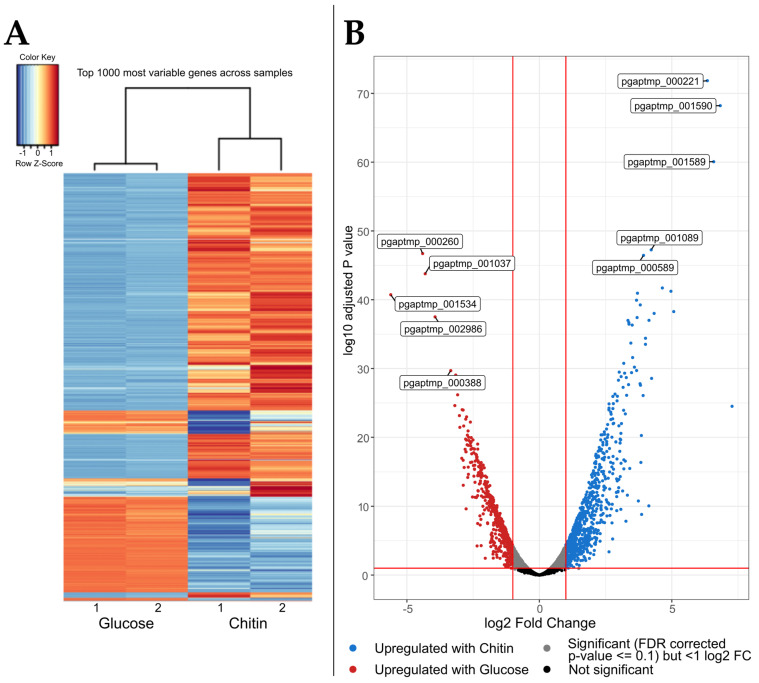
Differential transcriptomics results of *Jeongeupia wiesaeckerbachi* supplied with glucose or chitin. (**A**) Heatmap of the top 1000 most variable transcripts detected in duplicates, (**B**) Volcano plot of the differential transcriptomic dataset. The respective top 5 most significantly transcribed genes in glucose (red dots) or chitin (blue dots) media are labelled with their gene accession number.

**Figure 5 marinedrugs-21-00448-f005:**
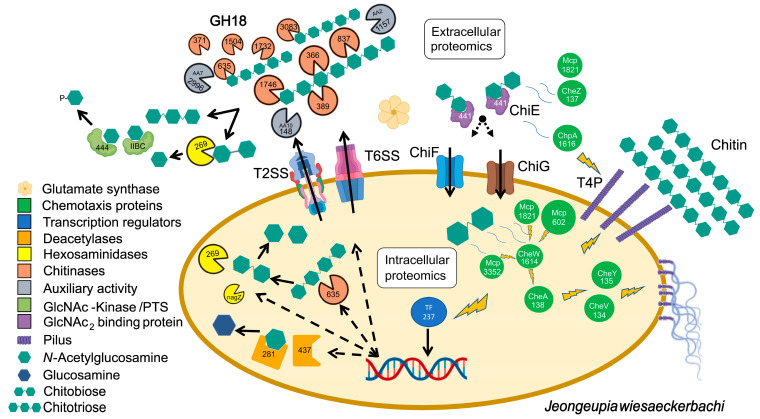
Schematic illustration of the proposed chitinolytic machinery of *Jeongeupia wiesaeckerbachi*. Extracellular proteins, depicted outside the bacteria cell, were detected in all triplicates cultivated in minimal media with colloidal chitin and a single sample cultivated with unbleached, decalcified crab shells. Intracellular proteins, illustrated inside the bacteria cell, were differentially upregulated with chitin minimal media compared to on glucose cultivated cells. Lightning bolts indicate signal transduction. Larger symbols indicate a higher significance or fold-change compared to enzymes of the same classification. Figure created with BioRender and Inkscape.

**Table 1 marinedrugs-21-00448-t001:** Top 10 most significant extracellular proteins on average, detected in all four samples with chitin as exclusive carbon and nitrogen source. Additionally, chitinoplastic enzymes among the commonly secreted proteins (386 in total) are provided with their respective significance rank in the lower half of the table.

Rank	Gene ID	Significance Score −10logP (Average)	Annotation (PGAP and dbCAN 3.0)	Complementary Annotation (KO)
**1**	pgaptmp_000837	631.37	glycosyl hydrolase family 18 protein	E3.2.1.14; chitinase [EC:3.2.1.14]
**2**	pgaptmp_001746	602.58	glycosyl hydrolase family 18 protein	E3.2.1.14; chitinase [EC:3.2.1.14]
**3**	pgaptmp_002871	595.18	branched-chain amino acid ABC transporter substrate-binding protein	livK; branched-chain amino acid transport system substrate-binding protein
**4**	pgaptmp_001064	582.92	TonB-dependent receptor	xylulose-5-phosphate/fructose-6-phosphate phosphoketolase
**5**	pgaptmp_000389	582.37	glycosyl hydrolase family 18 protein	E3.2.1.14; chitinase [EC:3.2.1.14]
**6**	pgaptmp_000021	560.63	MBL fold metallo-hydrolase	sdsA1; linear primary-alkylsulfatase [EC:3.1.6.21]
**7**	pgaptmp_002723	550.72	TonB-dependent siderophore receptor	TC.FEV.OM; iron complex outermembrane recepter protein
**8**	pgaptmp_000441	548.21	sugar ABC transporter substrate-binding protein	chiE; putative chitobiose transport system substrate-binding protein
**9**	pgaptmp_001471	525.48	porin	NA
**10**	pgaptmp_000366	523.85	glycosyl hydrolase family 18 protein	E3.2.1.14; chitinase [EC:3.2.1.14]
**49**	pgaptmp_002996	394,47	FAD-binding oxidoreductase (AA7)	NA
**56**	pgaptmp_001732	385.83	glycosyl hydrolase family 18 protein	E3.2.1.14; chitinase [EC:3.2.1.14]
**57**	pgaptmp_000635	384.40	glycosyl hydrolase family 18 protein	E3.2.1.14; chitinase [EC:3.2.1.14]
**58**	pgaptmp_000444	383.76	ATPase	gspK; glucosamine kinase [EC:2.7.1.8]
**91**	pgaptmp_003083	326.35	glycosyl hydrolase family 18 protein	E3.2.1.14; chitinase [EC:3.2.1.14]
**107**	pgaptmp_000269	308.55	carbohydrate-binding domain-containing protein (GH20)	HEXA_B; hexosaminidase [EC:3.2.1.52]
**134**	pgaptmp_000148	284.47	lytic polysaccharide monooxygenase	cpbD; chitin-binding protein
**198**	Pgaptmp_000371	238.27	glycosyl hydrolase family 18 protein	NA

**Table 2 marinedrugs-21-00448-t002:** Top 5 most significantly and top 5 most variably expressed proteins in the intracellular differential proteomics analysis of *Jeongeupia wiesaeckerbachi* with chitin as sole carbon and nitrogen source. Additionally, glucosamine metabolism-related proteins, which were upregulated in chitin medium are listed. Refer to Heatmap and volcano plot for a more holistic view of the data (Figure 2).

Gene ID	Significance	Log2 Fold Change	Annotation (PGAP and dbCAN3.0)	Complementary Annotation (KO)
pgaptmp_000442	67.67	11.11	sugar ABC transporter substrate-binding protein	chiE; putative chitobiose transport system substrate-binding protein
pgaptmp_000441	65.26	11.11	sugar ABC transporter substrate-binding protein	chiE; putative chitobiose transport system substrate-binding protein
pgaptmp_000337	64.6	50	nitrate reductase subunit beta	Identical
pgaptmp_003318	63.33	10	PrkA family serine protein kinase	identical
pgaptmp_000602	62.16	14.29	methyl-accepting chemotaxis protein	identical
pgaptmp_001215	49.26	50	SDR family oxidoreductase	NA
pgaptmp_002582	41.03	50	inclusion body family protein	aidA; nematocidal protein AidA
pgaptmp_001847	32.45	50	substrate-binding domain-containing protein	rbsB; ribose transport system substrate-binding protein
ggaptmp_000878	30.95	50	hypothetical protein	NA
pgaptmp_000237	39	33.33	hypothetical protein	NA
pgaptmp_000269	47.03	14.29	carbohydate-binding domain-containing protein (GH20)	HEXA_B; hexosaminidase [EC:3.2.1.52]
pgaptmp_000439	29.33	11.11	carbohydrate ABC transporter permease	chiG; putative chitobiose transport system permease protein
pgaptmp_000437	25.84	10	polysaccharide deacetylase family protein	pgdA; peptidoglycan-*N*-acetylglucosamine deacetylase [EC:3.5.1.104]
pgaptmp_000281	41.14	8.33	polysaccharide deacetylase family protein	NA
pgaptmp_001323	29.95	8.33	beta-*N*-acetylhexosaminidase (GH3)	nagZ; beta-*N*-acetylhexosaminidase [EC:3.2.1.52]
pgaptmp_000635	60.05	5.56	glycoside hydrolase family 18 protein	chitinase [EC:3.2.1.14]
pgaptmp_002871	59.65	5.56	branched-chain amino acid ABC transporter substrate-binding protein	identical
pgaptmp_000440	39.29	5.26	Sugar ABC transporter permease	chiF; putative chitobiose transport system permease protein
pgaptmp_003368	4.62	1.30	*N*-acetylglucosamine-specific PTS transporter subunit IIBC	nagE; *N*-acetylglucosamine PTS system EIICBA or EIICB component [EC:2.7.1.193]

**Table 3 marinedrugs-21-00448-t003:** Top 5 most variable transcripts of *Jeongeupia wiesaeckerbachi* in minimal media with colloidal chitin as exclusive carbon source. Transcripts were sorted according to the five lowest FDR (false discovery rate) corrected *p*-values. Additionally, all detected glycosyl hydrolase genes, which were upregulated under chitin media conditions, are listed. Refer to Appendix A for more information on differentially upregulated glucose transcripts.

Carbon Source	Rank	Adjusted *p*-Value	log2 Fold Change	Gene ID	Annotation (PGAP and dbCAN3.0)
Chitin	1	1.44 × 10^−72^	6.34	pgaptmp_000221	NirD/YgiW/YdeI family stress tolerance protein
	2	6.13 × 10^−69^	6.83	pgaptmp_001590	Flp family type IVb pilin
	3	8.63 × 10^−61^	6.58	pgaptmp_001589	Flp family type IVb pilin
	4	5.63 × 10^−48^	4.22	pgaptmp_001089	FtsX-like permease family protein
	5	3.56 × 10^−47^	3.93	pgaptmp_000589	hypothetical protein
Chitin	26	2.02 × 10^−34^	3.45	pgaptmp_000680	peptidoglycan-binding protein (GH19)
	52	1.15 × 10^−26^	2.77	pgaptmp_000371	glycosyl hydrolase family 18 protein
	102	4.18 × 10^−20^	2.54	pgaptmp_000306	carbohydate-binding domain-containing protein (GH20)
	294	8.38 × 10^−13^	1.80	pgaptmp_000836	glycosyl hydrolase family 18 protein
	309	2.1 × 10^−12^	1.90	pgaptmp_001841	glycosyl hydrolase family 18 protein
	348	1.08 × 10^−11^	2.25	pgaptmp_002137	glycosyl hydrolase family 18 protein
	448	4.33 × 10^−10^	2.92	pgaptmp_000372	glycosyl hydrolase family 18 protein
	453	4.9 × 10^−10^	1.66	pgaptmp_000302	chitinase (GH19)
	678	2.83 × 10^−7^	1.32	pgaptmp_001746	glycosyl hydrolase family 18 protein

## Data Availability

All data are provided in full in the results section, the Appendix B, or Appendix A of this paper. All raw datasets generated and/or analyzed during the current study are publicly available in the Zenodo online repository with the https://doi.org/10.5281/zenodo.8184099.

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
