# Peer review of "Proteomic and Transcriptomic Analyses to Decipher the Chitinolytic Response of Jeongeupia spp."

_marinedrugs, 2023, doi:10.3390/md21080448_

Round 1

Reviewer 1 Report

In this study, the authors performed proteomic and transcriptomic analyses of chitin utilization in Jeongeupia spp. Although there have been many studies on bacterial and fungal extracellular chitin-degrading enzymes, this study is very valuable in that it elucidates the relationship between these extracellular enzymes and the intracellular proteins that regulate them.

However, there are many mistakes that should be corrected. For example, Table 1 is listed three times, and in many places there are no references cited.

Others, for example,

(1) Line 90, “J. Wiesaecherbachi”: “J. wiesaecherhachi

(2) Line 131, “(     )”

(3) Lines 149, etc., “chitinoplastic enzymes”: It is unclear what this term means. Chitin-related enzymes?

(4) Line 480, “Chemicals & Consumables”: “Chemicals and Consumables”

(5) Line 494, “H2Odd”: double-distilled water?

(6) Line 497, “dest. water”: distilled water?

(7) Line 522, “(NH4)2Cl2”: “NH4Cl”

Author Response

We would like to thank the reviewer for their extensive efforts to improve the quality of our manuscript. Please find our point-by-point answers (A) to their questions (Q) in the following:

Q1:

However, there are many mistakes that should be corrected. For example, Table 1 is listed three times, and in many places there are no references cited.

A1:

We thank the reviewer for pointing out issues with the formatting, which originated in malfunctioning internal document links, showing the actual table instead of the referenced object’s title. We therefore replaced all internal reference links with ordinary text to eliminate said issue.

With reference to the reviewer’s comments on the referencing we would like to point out, that we have rechecked every reference with respect to the statements to be supported within the manuscript. We are sure that we provide a solid reference base for all statements made.

Could the reviewer’s impression have possibly been perceived because multiple citations were placed at the end of phrases? If so, we would like to point out that this was carried out not to interrupt the flow of a given sentence. Furthermore, if a statement from a literature source was continued upon, we would not repeatedly provide the very same citation again in every following sentence for reasons of redundance.

If it was, in fact, an issue with the formatting of references, we are happy to adjust the latter to eliminate misunderstandings and improve transparency.

If the reviewer is of the opinion that information from other sources was not appropriately provided, we are happy to add relevant references if the reviewer provides precise lines or phrases to address this severe issue.

Q2:

Others, for example,

(1) Line 90, “J. Wiesaecherbachi”: “J. wiesaecherhachi

(2) Line 131, “(     )”

(3) Lines 149, etc., “chitinoplastic enzymes”: It is unclear what this term means. Chitin-related enzymes?

(4) Line 480, “Chemicals & Consumables”: “Chemicals and Consumables”

(5) Line 494, “H2Odd”: double-distilled water?

(6) Line 497, “dest. water”: distilled water?

(7) Line 522, “(NH4)2Cl2”: “NH4Cl”

A2:

We thank the reviewer for the thorough reading of our manuscript, finding these errors in the process. We fixed all the listed typos (1) and clarified/altered the expressions (4 – 7). (7) is now in line 533, instead of 522.

Point (2) derived from issues with the internal reference function of Word, in addition to the redundant table copies, addressed by the reviewer in Q1.

Regarding (3), we changed the expression “chitin-enacting” to the more commonly utilized “chitin-active”. Additionally, brief introductions to the terms “chitinoplastic” and “chitin-active” were included the first time they are mentioned in the manuscript, respectively, to clarify their implications:

  1. 73: … chitin-active (interacting with chitin or COS molecules) enzymes were identified …
  2. 148: Other chitinoplastic (chitin structure altering) or chitin-enacting enzymes …

All aforementioned changes are highlighted in the revised manuscript for traceability.

Reviewer 2 Report

In this work, proteomic and transcriptomic analyses were performed to investigate the repertoire of genes and proteins that are involved in chitin utilization by the Gram-negative bacteria Jeongeupia wiesaeckerbachi. The work is suitable for publication in Marine Drugs, but there are several major and minor issues that should be addressed.

Major comments

1. Besides the enzymes that are obviously involved in chitin utilization, other proteins (branched-chain amino acid ABC transporter substrate-binding protein; TonB-dependent receptor; MBL fold metallo-hydrolase; TonB-dependent siderophore receptor; sugar ABC transporter substrate-binding protein; and porin) were present in the top 10 most significant extracellular proteins of J. wiesaeckerbachi cells grown in culture medium containing chitin as C and N source. Some points about these findings need more clear explanations. Were these proteins predicted to have any signal peptide or to be exported/secreted by non-classical pathways? Could the presence of these proteins in the secretome be regarded as a false positive result due to cell lysis? If not, what would be the functions of these proteins during chitin utilization?

2. The transcriptomic analysis was performed in cells grown on minimal colloidal chitin medium for three days, which explains the relatively low fold changes and adjusted P-value rankings of J. wiesaeckerbachi’s mRNAs coding for enzymes involved in chitin utilization. Besides the conclusion that the chitin-metabolism transcript upregulation is time-dependent, what relevant information can be obtained regarding chitin utilization genes from this transcriptomic analysis? Is there any new and significant data in relation to previous studies that is revealed by the transcriptomic analysis undertake in this work?

3. The genus Jeongeupia has been defined as a “chitinase-rich genera” (line 459). How did you come to this definition? How could you define a species or genera as a chitinase-rich taxon or as a chitinase-poor taxon?

4. The section Conclusions is too long (lines 639 to 726), and as such, it looks like more an extension of the sections Results and Discussion than the concluding remarks about the work.

Minor comments

1. Line 14. I suggest using the word “Gram” with an initial capital letter (be consistent throughout the text).

2. Line 69. The reference 19A [Arnold, N.D.; Garbe, D.; Brück, T.B. Isolation, Biochemical Characterization, and Sequencing of Two High-Quality Genomes of a Novel Chitinolytic Jeongeupia Species. Microbiologyopen 2023] is incompete. Please also verify the spelling of the journal name.

3. Line 73. What does “chitin-enacting enzymes” mean?

4. There are two extra copies of Table 1: one at pages 3-4 and another one at page 5.

5. Line 170. Verify the word “α-chitin”.

6. Line 172. Verify the word “SgLMPO10F”.

7. Lines 145, 149, 187, 202, 377, 433, and 679. What does “chytinoplastic” mean? Please check the spelling of this word and replace it with the correct term.

8. Line 159. I suggest using “−10logP” (not “−10lgP”) to represent the peptide-spectrum match (PSM) score, to avoid confusion with the natural logarithm (ln). Please be consistent throughout the text.

9. Line 325. What does “laxer threshold” mean?

10. Line 494. What does “H2Odd” mean?

11. Line 497. What does “dest. water” mean?

12. Line 498. I suggest using “10,000” instead of “10.000”.

13. Line 542. Change “was” (after 15 μL) to “were”.

14. In the section “3.4.3. Tryptic In-Gel Digestion and LC-MS/MS Analysis”, there are some information missing. Please carefully revise it and provide the missing information, like for example, the method used to perform tryptic in-gel digestion of proteins resolved by SDS-PAGE.

15. At the beginning of the second paragraph of the section “3.4.3. Tryptic In-Gel Digestion and LC-MS/MS Analysis”, an introductory phrase or sentence would be interesting, like “tryptic peptides from proteins resolved by SDS-PAGE were separated by reversed-phase chromatography as next described”.

16. Line 686. What does “chitin-enacting proteins” mean?

17. Line 688. For sake of more accuracy and clarity, I suggest using “penta-N-acetyl-chitopentaose”, which is a pentamer of N-acetylglucosamine, instead of chitopentaose.

Author Response

We would like to thank the reviewer for their extensive efforts to improve the quality of our manuscript, providing many valid and valuable points having to be addressed prior to publication. Please find our point-by-point answers (A) to their questions (Q) in the following:

Q1:

Besides the enzymes, that are obviously involved in chitin utilization, other proteins (branched-chain amino acid ABC transporter substrate-binding protein; TonB-dependent receptor; MBL fold metallo-hydrolase; TonB-dependent siderophore receptor; sugar ABC transporter substrate-binding protein; and porin) were present in the top 10 most significant extracellular proteins of J. wiesaeckerbachi cells grown in culture medium containing chitin as C and N source.

Some points about these findings need more clear explanations. Were these proteins predicted to have any signal peptide or to be exported/secreted by non-classical pathways? Could the presence of these proteins in the secretome be regarded as a false positive result due to cell lysis? If not, what would be the functions of these proteins during chitin utilization?

A1:

We thank the reviewer for their comment. The question which translocation pathways were responsible for export of these proteins, or whether they are among the false-positives due to cell lysis, is pivotal for the assessment of our data validity and must be clarified in the manuscript. Due to the reviewers exact observation, we investigated this problem and added the results in lines 250-268 as follows:

“We then investigated whether the top 10 most significant proteins found in the secretome were real hits or false positives due to cell lysis events. Deploying the SignalP 6.0 results for classical secretion pathways and our SignalP 6.0 corrected SecretomeP 2.0 prediction results for non-classical secretion pathways (Figure 1), we concluded that all detected extracellular proteins were real hits. Especially the five non chitin utilization associated proteins comprising of a BCAA-ABC-SBP (gene ID 2871), two TonB-dependent (siderophore) receptors (gene IDs 1064 & 2723), a class B metal beta-lactamase (MBL) fold metallo-hydrolase (gene ID 21) and a porin (gene ID 1471) had to be verified to confirm the validity of the dataset. According to SignalP 6.0 [28], four out of these five proteins are predicted to be secreted by means of the classical SEC pathway. To this end, three proteins (gene IDs 21, 1064 & 1471) were anticipated to be guided outside the bacterial cell with a signal peptide of type I, except for the BCAA-ABC-SBP, being directed by a lipoprotein signal peptide of type II. In contrast, the TonB-dependent siderophore receptor (gene ID 2723) was predicted by SecretomeP 2.0 [48] to be exported non-conventionally. Of the five chitin utilization associated proteins, including four glycosyl hydrolase family 18 proteins (gene IDs 366, 389, 837 & 1746) and the sugar ABC transporter substrate-binding protein (gene ID 441), all proteins were predicted to be exported classically with a signal peptide of type I, except for the non-classically exported GH18 (gene ID 1746), as mentioned above.”

Regarding the second part of the reviewer’s question, we speculated on possible functions of these proteins during chitin utilization in the following lines 188 – 203 based on available characterization studies and our in-silico predictions:

“Among these, the porin’s function as outer membrane channel is the most obvious. Knock-out experiments would have to show, which (potentially chitinolytic) enzymes would not be present in the secretome anymore and thus be transported through that specific porin. According to domain analysis with InterProScan [41,42], the two as TonB dependent receptors annotated proteins exhibit large β-barrel domains and might represent ligand gated channels or porins, thus serving as secretion facilitators. The BCAA-ABC-SBP is predicted to have high similarity to the Ile/Leu/Val-binding ABC transporter subunit. It might be upregulated and secreted due to the presence of amido-residues in the environment (media). Unspecific binding to, and import of, N’N’-diacetylchitobiose is possible [16,43], but a role in chemotaxis, pathogenicity, export and surface motility cannot be eliminated entirely for the superfamily of ABC transporters [44–46]. The involvement of the MBL fold-metallo hydrolase in chitinoplastic activities would have to be studied with knockout or expression experiments since typical functions of this superfamily comprise totally different hydrolytic activities such as alkylsulfatase, as suggested by KEGG annotation and InterProScan results (Table 1).”

Q2:

The transcriptomic analysis was performed in cells grown on minimal colloidal chitin medium for three days, which explains the relatively low fold changes and adjusted P-value rankings of J. wiesaeckerbachi’s mRNAs coding for enzymes involved in chitin utilization. Besides the conclusion that the chitin-metabolism transcript upregulation is time-dependent, what relevant information can be obtained regarding chitin utilization genes from this transcriptomic analysis? Is there any new and significant data in relation to previous studies that is revealed by the transcriptomic analysis undertake in this work?

A2:

We agree with the reviewer, that the time-dependent upregulation of chitin hydrolysis related genes is the main take away of the transcriptomic investigation in our study, which we tried to emphasize. It is only the second publication to our knowledge, that included proteomic investigation in correlation to transcript levels. Due to the difference in sampling times with diverging results, providing insights into the regulatory complexity of chitin utilization. We realize that results obtained by transcriptomics and proteomics are not directly comparable as they represent different time-constraints of the conducted measurements. Hence, we consider the proteomics data, which conventionally represents a kind of steady state look at cellular processes, to be most physiologically relevant, which we tried to convey in the lines 427 – 429. To enable a more correlated view of transcriptomic and proteomic data we have chosen time resolved transcriptomics particularly in the late stationary phase, where cells have reached maximal metabolic capacity.

This would also be the very first time, late stationary phase transcript levels of chitin utilizing bacteria is published, highlighting the importance of pili-mediated chemotaxis and chitin utilization transcript homeostasis at that point in time, which has not been demonstrated before and might be relevant to researchers in the field.

Q3:

The genus Jeongeupia has been defined as a “chitinase-rich genera” (line 459). How did you come to this definition? How could you define a species or genera as a chitinase-rich taxon or as a chitinase-poor taxon?

A3:

We thank the reviewer for pointing us to this inadequately explained statement. We came to this conclusion based on the study from Bai et al. published in 2014, where they compared genomes of terrestrial and aquatic bacteria regarding their chitinolytic enzyme systems (family 18 and 19 glycoside hydrolases (GH18 and GH19)):

Bai Y, Eijsink VG, Kielak AM, van Veen JA, de Boer W. Genomic comparison of chitinolytic enzyme systems from terrestrial and aquatic bacteria. Environ Microbiol. 2016 Jan;18(1):38-49. doi: 10.1111/1462-2920.12545. Epub 2014 Jul 15. PMID: 24947206.

Their analysis demonstrated, that on average, terrestrial bacteria possess 3.95 GH18 per genome, which is slightly skewed by Actinobacteria (5.38) and especially Streptomyces (5.82). Additionally, they could also identify only three species of Streptomyces with three GH19, with the average being 1.07 for Actinobacteria and a maximum of 2 for terrestrial bacteria in δ-Protobacteria.

With 13 GH18 and 3 GH19, our investigated Jeongeupia bacterium of the β-Protobacteria, belonging to the order of Neisseriales, possesses an unusually high number of chitinolytic enzymes in comparison to most other bacterial genomes.

We discussed this in depth in our previous publication and cited it in addition to the publication of Bai et al. in line 459 (now L. 472) to provide the appropriate sources for our definition.

Q4:

The section Conclusions is too long (lines 639 to 726), and as such, it looks like more an extension of the sections Results and Discussion than the concluding remarks about the work.

A4:

We agree with the reviewer, that our Conclusions section is in fact an extension of the Results and Discussion part, and therefore unusually long.

We did want to present a holistic view of all presented datasets, with resulting conclusions that could be drawn through this perspective in contrast to simply summarizing the manuscript. In our opinion, this would not have fit into any of the respective individual dataset’s sections.

If the reviewer does not approve of this manuscript structure, we are willing to move our current Conclusions section into a Results and Discussion section “2.4” and provide a conventional summary-style Conclusion.

Q5:

Line 14. I suggest using the word “Gram” with an initial capital letter (be consistent throughout the text).

A5:

We consistently changed the writing of Gram-negative to an initial capital letter throughout the text.

Q6:

Line 69. The reference 19A [Arnold, N.D.; Garbe, D.; Brück, T.B. Isolation, Biochemical Characterization, and Sequencing of Two High-Quality Genomes of a Novel Chitinolytic Jeongeupia Species. Microbiologyopen 2023] is incompete. Please also verify the spelling of the journal name.

A6:

The reference is a placeholder, as openly communicated with the editor since the manuscript is in the final stages of publication currently (accepted as of July 27th). We now received the DOI (10.1002/mbo3.1372) and will correct the reference as soon as possible. The reviewer is correct, that the spelling should be MicrobiologyOpen, instead, an error introduced by our citing software, hopefully eradicated by the official citation, once published.

Q7:

Line 73. What does “chitin-enacting enzymes” mean?

and

Line 686. What does “chitin-enacting proteins” mean?

A7:

We changed the expression “chitin-enacting” to the more commonly utilized “chitin-active” and briefly introduced it the first time it is mentioned in the manuscript, to clarify its implications:

  1. 73: … chitin-active (interacting with chitin or COS molecules) enzymes were identified …

(The other two instances are in lines 148 and 701).

Q8:

There are two extra copies of Table 1: one at pages 3-4 and another one at page 5.

A8:

We thank the reviewer for pointing out issues with the formatting, which originated in malfunctioning internal document links, showing the actual table instead of the referenced object’s title. We therefore replaced all internal reference links with ordinary text.

Q9:

Line 170. Verify the word “α-chitin”.

A9:

We thank the reviewer for pointing us to this phrase, which had a wrong citation, which we adjusted accordingly (Nakagawa et al. 2015 instead of Nakagawa et al. 2013).

We verify that α-chitin is employed correctly in this instance, describing the more stable and therefore crystalline chitin isoform, based on its antiparallel polysaccharide-chain alignment compared to the parallel alignment in β-chitin.

Based on your comment, we expanded the phrase in L. 170 to “…levels on the more stable and crystalline α-chitin…” to clarify the term.

Q10:

Line 172. Verify the word “SgLMPO10F”.

A10:

SgLMPO10F is the correct name of the LMPO of Streptomyces griseus (Sg) of the auxiliary family 10 utilized in the cited study from Nakagawa et al. (2015).

Nakagawa YS, Kudo M, Loose JS, Ishikawa T, Totani K, Eijsink VG, Vaaje-Kolstad G. A small lytic polysaccharide monooxygenase from Streptomyces griseus targeting α- and β-chitin. FEBS J. 2015 Mar;282(6):1065-79. doi: 10.1111/febs.13203. Epub 2015 Feb 4. PMID: 25605134.

Q11:

Lines 145, 149, 187, 202, 377, 433, and 679. What does “chytinoplastic” mean? Please check the spelling of this word and replace it with the correct term.

A11:

We added an introduction of the term “chitinoplastic” the first time it was mentioned in the manuscript to elaborate on the concept:

  1. 148: Other chitinoplastic (chitin structure altering) or chitin-enacting enzymes …

This includes not only chitinolytic (β-1,4-glycosidic bond cleaving), but also (de-)acetylating, transglycosylating or oxygenating enzyme activities targeting chitin or chitooligosaccharides.

We realize that this is no commonly utilized word but felt it to be impractical to circumscribe the concept whenever it was relevant in the manuscript. If the reviewer disagrees with this notion or has a more apt expression, we are open to changes.

Q12:

Line 159. I suggest using “−10logP” (not “−10lgP”) to represent the peptide-spectrum match (PSM) score, to avoid confusion with the natural logarithm (ln). Please be consistent throughout the text.

A12:

Based on the reviewer’s comment, we exchanged -10lgP with -10logP throughout the manuscript and supplementary material to avoid confusion with the natural logarithm ln.

Q13:

Line 325. What does “laxer threshold” mean?

A13:

We mean a less strict, or looser threshold, considering or defining all proteins with a 20% increase instead of 100% as upregulated, as we elaborate in the sentence (L.332) to demonstrate the influence of statistical differences in evaluation:

“If the dataset were evaluated with a laxer threshold, considering every intracellular protein to be upregulated 1.2-fold (or 20%) instead of 2-fold (or 100%) for example, a total of 257 proteins instead of 203 could be considered.”

Q14:

Line 494. What does “H2Odd” mean?

Line 497. What does “dest. water” mean?

A14:

We thank the reviewer for bringing our attention to these faulty expressions and replaced both with the accurate term diH2O (deionized water).

Q15:

Line 498. I suggest using “10,000” instead of “10.000”.

Line 542. Change “was” (after 15 μL) to “were”.

A15:

We fixed both semantical errors.

Q16:

In the section “3.4.3. Tryptic In-Gel Digestion and LC-MS/MS Analysis”, there are some information missing. Please carefully revise it and provide the missing information, like for example, the method used to perform tryptic in-gel digestion of proteins resolved by SDS-PAGE.

A16:

Dear reviewer, we referred to the references of Fuchs et al. and Engelhart-Straub & Cavelius for the full-length protocol of the tryptic in-gel digestion. The protocol is well established and frankly too extensive to be in the research manuscript.

We fully understand your sentiment though, that it can be tedious to be referred to other literature for protocols indefinitely. Therefore, we added the full-length protocol of the tryptic in-gel digestion and timsTOF LC-MS/MS sample preparation into the supplementary materials.

Q17:

At the beginning of the second paragraph of the section “3.4.3. Tryptic In-Gel Digestion and LC-MS/MS Analysis”, an introductory phrase or sentence would be interesting, like “tryptic peptides from proteins resolved by SDS-PAGE were separated by reversed-phase chromatography as next described”.

A17:

We agree with the sentiment, that a brief introduction improves the comprehensibility of the method. Therefore, we added the following in lines 574 – 576:

Extracted proteins from whole-cells were resolved by SDS-PAGE and subsequently digested with trypsin. Resulting peptides were then separated by reversed-phase chromatography and detected with a mass spectrometer as next described.

Q19:

Line 688. For sake of more accuracy and clarity, I suggest using “penta-N-acetyl-chitopentaose”, which is a pentamer of N-acetylglucosamine, instead of chitopentaose.

A19:

We agree with the reviewer and thank for the suggestion. The term chitopentaose was exchanged for the more accurate expression penta-N-acetyl-chitopentaose.

All aforementioned changes are highlighted in the revised manuscript for traceability.

Round 2

Reviewer 2 Report

The manuscript has been thoroughly revised and all the issues raised by this reviewer were properly answered. I have only one additional comment on this work.

1. Presentation of Table 1 (table legend and table´s content) is not correct. 

Author Response

We would like to thank the reviewer for their extensive efforts to improve the quality of our manuscript. Please find our point-by-point answers (A) to their questions (Q) in the following:

Q: Presentation of Table 1 (table legend and table´s content) is not correct. 

A: Dear reviewer, we re-checked the latest submitted manuscript and both the legend and content of table 1 are coherent and illustrate what they are supposed to. Maybe it is an issue with the word-file. Please refer to the provided PDF-file and make sure that you find table 1's presentation satisfactory.
